# Encouraging Physical Activity during and after Pregnancy in the COVID-19 Era, and beyond

**DOI:** 10.3390/ijerph17197304

**Published:** 2020-10-07

**Authors:** Lou Atkinson, Marlize De Vivo, Louise Hayes, Kathryn R. Hesketh, Hayley Mills, James J. Newham, Ellinor K. Olander, Debbie M. Smith

**Affiliations:** 1School of Psychology, Aston University, Birmingham B4 7ET, UK; 2Perinatal Physical Activity Research Group (PPARG), School of Psychology and Life Sciences, Faculty of Science, Engineering and Social Sciences, Canterbury Christ Church University, Canterbury CT1 1QU, UK; marlize.devivo@canterbury.ac.uk (M.D.V.); hayley.mills@canterbury.ac.uk (H.M.); 3Population Health Sciences Institute (PHSI), Newcastle University, Newcastle upon Tyne NE1 7RU, UK; louise.hayes@ncl.ac.uk; 4UCL Great Ormond Street Institute of Child Health, London WC1N 1EH, UK; Kathryn.hesketh@ucl.ac.uk; 5Faculty of Health and Life Sciences, Northumbria University, Newcastle upon Tyne NE1 8QH, UK; james.newham@northumbria.ac.uk; 6Centre for Maternal and Child Health Research, School of Health Sciences, University of London, London WC1E 7HU, UK; Ellinor.olander@city.ac.uk; 7Manchester Centre for Health Psychology, University of Manchester, Manchester M13 9PL, UK; debbie.smith-2@manchester.ac.uk

**Keywords:** physical activity, pregnancy, postnatal, COVID-19, behaviour change, theory

## Abstract

Physical activity is known to decline during pregnancy and the postnatal period, yet physical activity is recommended during this time due to the significant health benefits for mothers and their offspring. As a result of the COVID-19 pandemic and the restrictions imposed to reduce infection rates, pregnant and postnatal women have experienced disruption not just to their daily lives but also to their pregnancy healthcare experience and their motherhood journey with their new infant. This has included substantial changes in how, when and why they have engaged with physical activity. While some of these changes undoubtedly increased the challenge of being sufficiently active as a pregnant or postnatal woman, they have also revealed new opportunities to reach and support women and their families. This commentary details these challenges and opportunities, and highlights how researchers and practitioners can, and arguably must, harness these short-term changes for long-term benefit. This includes a call for a fresh focus on how we can engage and support those individuals and groups who are both hardest hit by COVID-19 and have previously been under-represented and under-served by antenatal and postnatal physical activity research and interventions.

## 1. Introduction

In March 2020, pregnant women in the UK were identified as a vulnerable group and advised to self-isolate for 12 weeks as a precaution against infection from the novel Coronavirus SARS-CoV-2 which causes COVID-19. Following this advice involved reducing face-to-face social contact, staying at home and only leaving for essential reasons. Early research indicates a number of deleterious consequences of this isolation among some pregnant and postpartum women, including an increase in depression and anxiety, and reduced physical activity [1]. However, the same study also indicated that women who met guidelines of at least 150 mins/week of moderate intensity physical activity were significantly less depressed and anxious than those women who did not. Physical activity (PA) is well established among the general and maternal populations as having a protective effect on both mental and physical health. Evidence now also shows that physical inactivity is a risk factor for diagnosis, serious illness, and death with COVID-19 [2]. This presents a significant dilemma—how to promote and support PA for all as a means of reducing the risks of COVID-19, while protecting the vulnerable through social distancing and other measures that may limit opportunities to be active [3].

## 2. Physical Activity in Pregnancy and Postpartum

Despite significant efforts in recent years, there remains a marked decline in PA levels during and after pregnancy [4,5,6]. Importantly, a reduction in PA has also been observed among others in the family unit, extending from pregnancy into the postpartum period [7], demonstrating the wide reaching impact on PA of this major life event. A large number of studies have been conducted to understand the influences on PA in these populations (recently synthesised by Harrison et al. [8]), with a high degree of agreement between studies on the key determinants of behaviour and recommendations for intervention targets. For example, commonly reported barriers to PA in pregnancy include lack of time, motivation, confidence, knowledge and support; fatigue; safety concerns; and pregnancy discomforts [8], whereas barriers such as lack of motivation, support, and energy; social isolation; and lack of access to appropriate and affordable programs and facilities are synonymous with the postnatal period [9]. Whilst intrapersonal themes are consistently reported as both barriers and enablers to PA participation, variability in the influence of theory-based factors on PA behaviour throughout the stages of pregnancy (i.e., trimesters) and following childbirth have become apparent [10]. This suggests that flexible person-centred strategies and interventions are needed to accommodate the many changes associated with the perinatal period [8], and may indicate why trials of interventions to increase or sustain PA have rarely shown a significant positive effect in either pregnant [11] or postnatal women [12]. Recognizing, therefore, that one-size-fits-all approaches are insufficient, new approaches are required. This commentary proposes that the disturbance of our daily lives brought about by the COVID-19 pandemic provides us as researchers and practitioners with a chance to “disrupt” our field and use the new insights gained from this opportunity to include and support more women.

## 3. Challenges and Opportunities of COVID-19

### 3.1. Changes to Working and Lifestyle Patterns, and Increased “Online” Content

Across the globe, normal daily routines of work, education and home life have been severely disrupted by lockdown measures implemented to slow the spread of the novel coronavirus. Such measures have already been shown to have had both negative and positive impacts on lifestyles, including PA [13]. On one hand, closure of gyms and leisure facilities, less opportunities for active travel to work and school, and general limitations on movement have reduced opportunities for some exercise modalities. Closure of schools and childcare facilities may also have reduced time available for physical activities for parents. A recent survey by the Active Pregnancy Foundation of 445 pregnant and post-natal women indicated that 50% were doing less PA during lockdown than prior to it [14]. This compares to Sport England [13] data from the general adult population that showed that between 34% and 38% of all adults reported lower levels of PA during lockdown, suggesting that the pregnant and post-natal populations may have been disproportionately affected by these measures [13]. On the other hand, both the Active Pregnancy Foundation and Sport England data show that around a third of their participants reported higher levels of PA during lockdown than in the weeks prior.

Based on our existing evidence base regarding barriers and facilitators for PA during pregnancy and the post-natal period, we can identify where the potential lies for harnessing the opportunities of these disrupted lifestyles. For example, pregnant women consistently suggest that pregnancy-specific exercise classes can overcome their concerns regarding the safety of exercise [15]. However, these specialised sessions can be expensive and require participants to overcome the other well-established barriers of tiredness, work and childcare responsibilities, and to travel to a venue at a specific time. The rapid move to online workouts by the fitness industry during lockdown now provides women access to professional, pregnancy-specific exercise sessions, without leaving their home and families, at a time to suit them. The move to working from home may also afford some women the flexibility in their daily schedules to incorporate more PA, where previously they would have been too tired by the end of the day. Small-scale trials of home-based workouts have shown that postnatal women find these beneficial for both mental health and overcoming barriers to PA [16]. The accessibility and scalability of content via online platforms has the potential to facilitate much more widespread adoption of home-based PA during pregnancy, but key to this will be ensuring that the services on offer truly overcome established barriers. This means they must be flexible to access and participate in, free or low-cost, and delivered by qualified and relatable professionals. If content is expensive to access, requires high specification technology or additional fitness equipment in the home, lacks credibility and/or fails to engage in the delivery, it will likely fail to reach a wide audience and produce sustained PA adoption. To this end, governments must consider financially supporting the creation, distribution and accreditation of such resources as part of their maternity care and public health strategies.

Similarly, postnatal women can not only now access suitable workouts via the internet, many of which include their baby and so do not require childcare to participate in, but can connect with other mothers via apps and social media to provide valuable social interactions in addition to PA. As in pregnancy, online content for postnatal women must be provided by qualified and relatable professionals, be free or low-cost, and be flexible to access if it is to overcome existing barriers to PA. For example, the Active Pregnancy Foundation survey reported that 75% of pregnant and post-natal participants were seeking free resources online [14]. The phenomenal success of the Joe Wicks PE sessions in the UK, which provided a daily workout for children and their families and peaked at over 950,000 households tuning in on a single day (https://www.guinnessworldrecords.com/news/2020/4/joe-wicks-pe-with-joe-smashes-youtube-livestream-record-614934), also demonstrates the potential for whole families to regularly engage in PA together, using live streamed content. Whole family PA is likely to be vital to maintaining or increasing PA among the pregnant and post-natal women who report caring for other children as a barrier to their own PA. Additionally, both pregnant and post-natal women are more likely to be physically active if their partner is also active, and children of more active parents participate in more PA than those with less active parents [17]. Encouragingly, the Sport England data show that adults with children in the household have been more active during lockdown than those without [13]. Hence, whole-family PA can facilitate a more active post-COVID generation.

In addition to online-facilitated home-based PA opportunities, significant numbers of adults have participated in some of the simplest and most accessible activities: walking, cycling and informal play during lockdown [13]. While some reduction in these has been observed after sport and leisure facilities re-opened, among women and families with low incomes, living in remote areas, or with caring responsibilities, these remain some of the most feasible PA opportunities. Interventions that could support the continued engagement in these activities may include ensuring routes are safe and well-maintained, messaging via local authorities and health professionals about the benefits of these activities, and building support networks such as walking groups/meet up points and online communities. Similarly, while being isolated at home can be perceived to reduce PA opportunities, housework, gardening and active play with children can all contribute to the accumulation of recommended amounts of PA, and appropriate messaging regarding this could support the 52% of pregnant and post-natal women who reported worrying about leaving the house to exercise [14]. Messaging needs to be consistent with current guidance [18], from credible and relatable sources, and provide sufficient detail for women to carry out these activities safely.

### 3.2. Healthcare Professionals’ Role and Availability

Pregnant women consistently report that one of their main sources of PA information is healthcare professionals, due to their regular interactions and being seen as a credible source of information [19]. During COVID-19, these interactions have changed in a number of ways. Firstly, the information provided during the appointments now focuses on COVID-19, including changes to clinical care and birth choices, and ability to have their partner with them during labour [20]. Women’s anxiety and stress has been found to be heightened during the pandemic [21,22], so discussions may be more focused upon this than on lifestyle behaviours. Secondly, the number of appointments have decreased in some cases and the nature of appointments have changed. Appointments are offered online or via phone, with very few face-to-face meetings [20]. This is beneficial in many cases due to lower risk of infection, and women report being anxious about attending clinical settings such as hospitals [21]. That said, these appointments may be less conducive to discussing health-promoting behaviours such as PA. Finally, there have been changes to the workforce. There are several examples of healthcare professionals being redeployed into other areas of critical care [20,23]. Further, healthcare professionals also have a high risk of COVID-19 infection [24]. Taken together, there are fewer healthcare professionals available to care for pregnant and postnatal women and those available may have less time to discuss PA behaviour. Similarly, women may be reluctant to attend exercise sessions at gyms and sports facilities, where they can be advised by qualified fitness professionals, due to concerns about contracting the virus. While some cases of transmission linked to sport and fitness activities were reported early in the pandemic [25,26], more recent data suggest that since appropriate hygiene and social distancing measures have been implemented, few cases have been reported in gyms and fitness facilities [27]. Nevertheless, many women may prefer to take a precautionary approach [14].

Considering the above, other credible sources of information and advice need to be promoted during this time when healthcare professionals are occupied by other priorities, and exercise professionals are less accessible. The internet is often mentioned as a source of information by women [19,28] and could be harnessed to provide information that is not only reliable but accessible at times to suit women and their partners. This includes online campaigns such as the UK campaign This Mum Moves (https://www.babybuddyapp.co.uk/this-mum-moves). Prior to the current pandemic, antenatal healthcare professionals reported being motivated to promote PA to women but lacking the skills and resources to do this consistently [29]. Increased availability and acceptance of information from these alternative sources could both relieve the pressure on healthcare professionals and provide fruitful new avenues for PA intervention.

### 3.3. Physical Activity as a Coping Strategy for Mental Wellbeing

Under normal circumstances, pregnant women and new mothers report higher levels of stress and feelings of depression and anxiety in comparison to the general population [30]. Maternal stress and anxiety are well documented to have an adverse effect on prenatal and postnatal development of babies and early intervention is vital [31]. Research has shown that lockdown has significantly impacted the stress, anxiety and depression levels expressed by the UK public [32,33] and pregnant women [21]. Yet, as previously noted, Davenport et al. [1] reported that pregnant and post-natal women who met PA guidelines had lower levels of depression and anxiety, and 88% of women who participated in the Active Pregnancy Foundation [14] survey indicated that PA had helped them to manage their mental health. This suggests that the protective effect of PA on mental wellbeing has been important during the early stages of the pandemic, and is likely to be an important tool to help women in the “new normal”. Previous research has highlighted that women are more motivated by benefits for their baby over benefits for themselves [34], and thus the benefits of PA both for infant development and reducing the possible impacts of COVID-19 on women’s wellbeing could be promoted explicitly, to encourage further engagement in PA in this population. As always, it is important to be mindful of the pressures that many women already feel to be “good mothers” and “good pregnant women” by complying with explicit guidance and implicit societal expectations [19]. Careful consideration must be given to how messages around PA are framed and disseminated, to avoid creating guilt or stigma among women who may have very real challenges to becoming or staying active. However, the growing awareness of the mental and physical benefits of PA that has become apparent during the pandemic provides more opportunity than ever before to speak directly to women and their families about the multiple positive effects of an active lifestyle.

### 3.4. Health Inequalities

The current pandemic has brought health inequalities into particularly sharp focus. Existing inequalities in health, and its determinants, have contributed to the impact of COVID-19 being amplified in already disadvantaged populations [35]. Inequities in the impact of the virus amongst pregnant women have also been reported. For example, a prospective national-population-based cohort study conducted in the UK during the initial months of the pandemic revealed that a high proportion (56%) of pregnant women admitted to hospital with COVID-19 were from black or other minority ethnic groups and 70% of them had overweight or obesity [36].

As stated above, the restrictions associated with the COVID-19 pandemic, including isolation and country-wide “lockdowns”, provided impetus for the fitness industry to provide online workouts, tailored to specific groups, including pregnant women. These addressed several of the traditional barriers to pregnant women participating in PA, including women working around pregnancy symptoms, and existing childcare needs. However, it is imperative that the opportunities afforded by this shift to online classes and normalising home-based PA do not benefit some groups of women at the expense of the most vulnerable and already disadvantaged, in terms of participation in PA and its associated health benefits.

Intervention-generated inequalities, i.e., where effective interventions disproportionately benefit advantaged groups, are a risk of PA interventions that are not appropriately targeted at those with the greatest need [37]. As we move forward, we must ensure that research to evaluate the provision of, and access to, PA for pregnant women and the development of future interventions delivered at scale avoids this. Equally, it is imperative that those who are both less likely to engage in sufficient PA and are currently under-served by research are sufficiently represented and have meaningful impact on intervention development and evaluation [38]. Whilst many pregnant women are insufficiently active, regardless of demographic factors, failing to include pregnant women from those groups that already typically have low PA levels (i.e., those who are socio-economically disadvantaged, from Black Asian and Minority Ethnic (BAME) populations and who have overweight and obesity) will have wide-ranging negative consequences to both maternal and child health. Indeed, interventions will not only fail to narrow but are likely also to increase the gap in health inequalities if they continue to be designed and delivered for those who need them the least, while not reaching those who need them the most.

To date, BAME and low-income groups have not always been widely represented in research and intervention design. This may be because effective engagement can require long-term investments, such as developing relationships with community groups and including their expertise in the development of procedures, extensive formative phases and pilot testing. The need for longer project timelines and adequate community feedback may in part explain this engagement gap. Additional resources such as translation services, bilingual research staff, flexibility in data collections, culturally tailored resources and staff training can also significantly increase costs. The resources, time, expertise and funds required to be fully inclusive have not always been available across all research programmes [39].

### 3.5. Development of Theoretical Understanding and Intervention Components

While there is agreed understanding of the determinants of PA, and awareness of the groups of women less likely to achieve recommended levels of PA, interventions purposefully designed to address these barriers have been predominantly ineffective. There remains a disconnect between theoretical understanding and intervention success, and the stipulations of lockdown may provide a unique opportunity to better understand the relative importance and interplay of individual determinants of PA. For example, De Vivo et al.’s meta-analysis highlighted the importance of the individual construct of subjective norm (perceived pressure to conform to others’ expectations) on PA intentions and behaviours in pregnancy [40], and the importance of social normative behaviour has been an integral construct within several explanatory frameworks (Social Norms Theory, HAPA, Social Cognitive Theory). Under lockdown, we can more closely see the importance of other constructs such as attitudes and perceived behavioural control, as variation in in-person social interaction has essentially been “levelled” across the population. As physical and social environments change to reflect physical restrictions and behavioural guidance, the relative importance of theoretical constructs will also change and intervention components will be required to address different determinants than those previously targeted. COVID-19 may also represent a unique situation whereby the removal of certain environmental determinants fosters more sustainable patterns of behaviour that can be implemented post-lockdown, for example, building implementation intentions focused on home-based stimuli (e.g., “I will exercise when I wake up, in the living room”) that are less susceptible to external barriers (e.g., “I plan to go to the gym after work”—but is derailed by traffic).

Similarly, the Capability, Opportunity, Motivation and Behaviour (COM-B) model has been applied to understanding antenatal PA [41], with Social Opportunity, Physical Opportunity and Physical Capability being identified as key factors, as well as having the necessary knowledge about safe exercises to have the Psychological Capability to be active. During lockdowns, physical opportunity is impacted (swimming pools and leisure centres closed), as is social opportunity (social distancing and isolation reduce group exercise and being active with others), and this may impact perceived capability (pregnant women may see themselves as only safe when exercising with an instructor or undertaking low-impact exercise such as swimming). This allows more granular understanding of the importance of Motivation, as indicated by the reports of women being motivated towards PA as a coping strategy for mental wellbeing [14]. These emerging sources of motivation not only enable us to develop a deeper understanding of the complex determinants of PA during and after pregnancy, they can also provide an opportunity to develop and test innovative intervention strategies that may provide a much-needed breakthrough in improving PA rates in a population that continues to be predominantly inactive.

## 4. Moving the Field Forward

COVID-19 represents an opportunity to gain better understanding of the relative importance of theoretical determinants, develop and test new interventions and tap into new sources of motivation to promote and support PA in pregnant and post-natal women. However, before embarking on new streams of research, an important first step is to honestly investigate why we have thus far failed to engage some groups of women in research in this field. As highlighted above, additional resources and greater community engagement in research planning and development are likely to be needed. Alongside this, given the uncertainty of COVID-19, future interventions studies will inevitably be considering how they can be delivered remotely or by digital methods. Rather than simply replicating in a different modality, researchers and practitioners should consider how they can use such technology to better understand the populations they engage with, for example, using user analytics to understand when and how often participants engage with digital content. This provides indications of how country of origin and ethnicity may influence engagement, and in turn effectiveness. Alternatively, studies looking to use digital platforms to deliver PA interventions can use existing technology for geo-mapping of areas participants may run or walk in, to better understand how environment may influence PA. For example, are those living in deprived areas without green space to walk in less likely to engage in interventions as it is genuinely more difficult to adhere? However, it is important that any adaptations to digital platforms do not create or exacerbate health inequalities. The modalities available to deliver such interventions may not be readily accessible to everyone, and it is equally important to document the extent to which this can be a deterrent for uptake and inclusion.

Ultimately, if the aforementioned opportunities presented by the pandemic are to be adequately leveraged for long-term benefit, a rapid response from funders, researchers and publishers in the behavioural sciences will be required, akin to that seen in the field of biomedical science, to quickly develop treatments and vaccines for the virus. Cross-sectional surveys with multiple timepoints can provide a unique, interrupted time series with valuable information on the disruption caused by COVID-19 and how things change as restrictions lift or are reintroduced. However, it is also opportune to work with international collaborators at different stages of their respective lockdowns, to rapidly implement and test interventions that have already shown promise, to better understand how interventions may prove more or less effective depending on what determinants of behaviour are inhibited. For example, how does an app-based intervention work in a country where gyms are open compared to where they are not? Most importantly, the response from the research community needs to accept and support the need for interventions that address the problem above the level of individual women and health professionals.

## 5. Conclusions

Pregnancy and the postnatal period is recognized as posing unique challenges to being sufficiently active. The COVID-19 pandemic has presented additional difficulties through disrupting social contact and daily routines, and limiting physical opportunities for PA. However, these changes have also provided opportunities to look at this behaviour from different perspectives, and to deepen our understanding of the relative influences of physical, social and motivational factors. While preliminary work is already providing new insights, such as the increased importance of PA to managing mental health, more high-quality, international research is required to make the most these new opportunities.

Moreover, the additional burden of COVID-19 on the BAME community, socially deprived populations and individuals with obesity should provide an impetus for renewed focus on addressing the lower-than-average PA levels observed in these communities. Researchers and funders must be prepared to take on the task of conducting research that truly engages, represents and benefits those who currently experience significant health inequalities. Accepting that this will take more time and resources is vital, as is the recognition that we must be braver and more innovative in our approaches. Fortunately, the shifts in our daily lives brought about by the pandemic have also opened up new prospects for wider engagement and reach, with the greater acceptance and use of online content, less reliance on busy health professionals, and changes to work and home lives facilitating PA as a family. Our challenge now is to harness these short-term changes for long-term benefit, particularly among those communities with arguably the most to gain from a more active antenatal and postnatal lifestyle.

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
