# Peer review of "Encouraging Physical Activity during and after Pregnancy in the COVID-19 Era, and beyond"

_ijerph, 2020, doi:10.3390/ijerph17197304_

Round 1

Reviewer 1 Report

This manuscript presents very interesting topis these days, but I have some minor comments about it.

Dear Authors, follow the Instructions for Authors due to the way of presentation the references in text. Also modify the section References due to Instructions for Authors.

In the Acknowledgments are information about funds, but in above section there is no information about it.

Author Response

Reviewer comments:

Dear Authors, follow the Instructions for Authors due to the way of presentation the references in text. Also modify the section References due to Instructions for Authors.

Our response:

We have amended the referencing style in text and in the reference list, using the MDPI Endnote template. 

Reviewer comments:

In the Acknowledgments are information about funds, but in above section there is no information about it.

Our response:

The funding section has been amended as follows:

"Funding: No external funding was provided specifically for the preparation of this article. Authors belong to a research working group which has received funding to support its work, see Acknowledgements for details."

Reviewer 2 Report

Thanks for inviting me to review the manuscript ID ijerph-934213 entitled Encouraging Physical Activity during and after Pregnancy in the Covid-19 era, and beyond

This manuscript focused not only on the importance of physical activity during pregnancy and its consequences, but also on the lack of physical exercise associated with COVID-19 pandemic period. Therefore, Atkinson et al., suggested strategies to maintain physical activity during and after pregnancy in the digital era, and they addressed the importance of new studies that focus on using technology and environmental geo-mapping to understand how this time should be impacted and may influence physical activity. This reflection provided by the authors demonstrates how emotional health may arise in pregnant women, affecting mental health.

While reading this paper, I see that healthcare professionals have a high-risk COVID-19 infection, but I do not see that SARS-CoV-2 can spread at Gyms and Fitness Class. Authors should provide an analysis of negative impact in going to the gym during the COVID-19 pandemic:

  1. Emerg Infect Dis. 2020 Aug;26(8):1917-1920. doi: 10.3201/eid2608.200633
  2. Epidemiol Infect. 2020; 148: e120. doi: 10.1017/S0950268820001326

Author Response

Reviewer's comments:

While reading this paper, I see that healthcare professionals have a high-risk COVID-19 infection, but I do not see that SARS-CoV-2 can spread at Gyms and Fitness Class. Authors should provide an analysis of negative impact in going to the gym during the COVID-19 pandemic:

  1. Emerg Infect Dis. 2020 Aug;26(8):1917-1920. doi: 10.3201/eid2608.200633
  2. Epidemiol Infect. 2020; 148: e120. doi: 10.1017/S0950268820001326

Our response:

We have included the following to acknowledge the potential risks and the concerns of pregnant and postnatal women.

Line 157-162:

"Similarly, women may be reluctant to attend exercise sessions at gyms and sports facilities, where they can be advised by qualified fitness professionals, due to concerns about contracting the virus. While some cases of transmission linked to sport and fitness activities were reported early in the pandemic[25,26], more recent data suggest that since appropriate hygiene and social distancing measures have been implemented, few cases have been reported in gyms and fitness facilities[27]. Nevertheless, many women may prefer to take a precautionary approach[14]."

Reviewer 3 Report

Congratulations to the aurthors for this so needed article highlighting the importance of pre&postnatal physical activity during this pandemic time and the neccesity of overcoming barriers to make physical activity affordable and available for every woman.

Please find below my comments on your article.

Line 77-78: Please add a reference to the sentence starting: Such measure have already been shown...

Lines 94-125: Authors reach very important points within this section. However, I feel that it is not stressed enough the importance of seeking and finding qualified professionals when engaging in pre&postnatal activities, specially this is not mentioned within the postnatal section. Another important point is the neccesity of low-cost or free activities so they are affordable for everyone. Therefore, inequalities can be overcome. However, it is not mention how these activities could be supported or funded and the huge need of investment in successful physical activity strategies to keep promoting an active life-style during and after pregnancy. The last one appears further within the text, although this should be also mentioned in this section.

Line 254: When authors use the acronym COM-B, please add the whole term (Capacity, Opportunity, Motivation and Behavour model)

Along the text authors use the term "physical activity" or PA. Please keep consistency within the text when using it either physical activity or PA.

Good luck!

Author Response

Reviewer's comments:

Line 77-78: Please add a reference to the sentence starting: Such measure have already been shown...

Our response:

Reference has been added.

Reviewer's comments:

Lines 94-125: Authors reach very important points within this section. However, I feel that it is not stressed enough the importance of seeking and finding qualified professionals when engaging in pre&postnatal activities, specially this is not mentioned within the postnatal section. Another important point is the neccesity of low-cost or free activities so they are affordable for everyone. Therefore, inequalities can be overcome. However, it is not mention how these activities could be supported or funded and the huge need of investment in successful physical activity strategies to keep promoting an active life-style during and after pregnancy. The last one appears further within the text, although this should be also mentioned in this section.

Our response:

We absolutely agree with the reviewer regarding the need for activities for both pregnant and postnatal women to be provided by qualified professionals, and to be available at free or low cost. We have added emphasis on these points through the following amendments:

Lines 103-107:

"This means they must be flexible to access and participate in, free or low cost, and delivered by qualified and relatable professionals. If content is expensive to access, requires high specification technology or additional fitness equipment in the home, lacks credibility and/or fails to engage in the delivery, these will likely fail to reach a wide audience and produce sustained PA adoption.  To this end, governments must consider financially supporting the creation, distribution and accreditation of such resources, as part of their maternity care and public health strategies."  

Lines 110-112:

"As in pregnancy, online content for postnatal women must be provided by qualified and relatable professionals,  free or low cost, and flexible to access, if it is to overcome existing barriers to PA."

Reviewer's comments:

Line 254: When authors use the acronym COM-B, please add the whole term (Capacity, Opportunity, Motivation and Behavour model)

Our response:

Amended to full name of the model

Reviewer's comments:

Along the text authors use the term "physical activity" or PA. Please keep consistency within the text when using it either physical activity or PA.

Our response:

Amended to consistently use PA throughout the text